# Trigeminocardiac Reflex Induced by Maxillary Nerve Stimulation during Sphenopalatine Ganglion Implantation: A Case Series

**DOI:** 10.3390/brainsci10120973

**Published:** 2020-12-11

**Authors:** Yousef Hammad, Allison Mootz, Kevin Klein, John R. Zuniga

**Affiliations:** 1Department of Oral & Maxillofacial Surgery, The University of Texas Southwestern Medical Center, 5323 Harry Hines Blvd, Dallas, TX 75390, USA; yousef.hammad@utsouthwestern.edu; 2Department of Anesthesiology and Pain Management, The University of Texas Southwestern Medical Center, 5323 Harry Hines Blvd, Dallas, TX 75390, USA; 3Departments of Anesthesiology & Pain Management and Otolaryngology, Head & Neck Surgery, The University of Texas Southwestern Medical Center, 5323 Harry Hines Blvd, Dallas, TX 75390, USA; kevin.klein@utsouthwestern.edu; 4Departments of Oral & Maxillofacial Surgery and Neurology, The University of Texas Southwestern Medical Center, 5323 Harry Hines Blvd, Dallas, TX 75390, USA; john.zuniga@utsouthwestern.edu

**Keywords:** trigeminocardiac reflex, sphenopalatine ganglion, OMFS, anesthesia, multidisciplinary approach

## Abstract

Background: The trigeminocardiac reflex (TCR) is a brainstem reflex following stimulation of the trigeminal nerve, resulting in bradycardia, asystole and hypotension. It has been described in maxillofacial and craniofacial surgeries. This case series highlights TCR events occurring during sphenopalatine ganglion (SPJ) neurostimulator implantation as part of the Pathway CH-2 clinical trial “Sphenopalatine ganglion Stimulation for Treatment of Chronic Cluster Headache”. Methods: This is a case series discussing sphenopalatine ganglion neurostimulator implantation in the pterygopalatine fossa as treatment for intractable cluster headaches. Eight cases are discussed with three demonstrating TCR events. All cases received remifentanil and desflurane for anesthetic maintenance. Results: Each patient with a TCR event experienced severe bradycardia. In two cases, TCR resolved with removal of the introducer, while the third case’s TCR event resolved with both anticholinergic treatment and surgical stimulation cessation. Conclusion: Each TCR event occurred before stimulation of the fixed introducer device, suggesting the cause for the TCR events was mechanical in origin. Due to heightened concern for further TCR events, all subsequent cases had pre-anesthesia external pacing pads placed. Resolution can occur with cessation of surgical manipulation and/or anticholinergic treatment. Management of TCR events requires communication between surgical teams and anesthesia providers, especially during sphenopalatine ganglion implantation when maxillary nerve stimulation is possible.

## 1. Introduction

The trigeminocardiac reflex (TCR) is a well-established brainstem reflex that occurs following stimulation of the trigeminal nerve anywhere along its course and can result in bradycardia, asystole, hypotension, apnea, and gastric hypermobility [1]. The afferent limb of the oculocardiac reflex (OCR), a subtype of TCR, originates with afferent fibers of the long and short ciliary nerves that travel within the ophthalmic division of the trigeminal nerve [2]. This afferent limb continues to the gasserian ganglion and joins the main sensory nucleus of the trigeminal nerve. Internuncial fibers in the reticular formation connect the afferent limb of TCR to the efferent limb from the motor nucleus of the vagus nerve to the depressor fibers of the vagus nerve, which ends in the myocardium [2]. TCR can be defined as the onset of bradycardia lower than 60 beats per minute and a drop in mean arterial blood pressure of 20% or more from baseline due to intra-operative manipulation or traction of the trigeminal nerve [3]. TCR most commonly occurs intra-operatively following stimulation of the afferent fibers of the trigeminal nerve, and it is usually identified quickly via monitoring of hemodynamic changes. In the vast majority of cases, hemodynamic parameters return to normal levels without further occurrences of TCR following withdrawal or removal of the stimulation and/or manipulation of the trigeminal nerve.

TCR has been reported during various surgeries of the head and neck involving the distribution of the trigeminal nerve where manipulation of the trigeminal nerve/ganglion may occur. The afferent pathway of TCR can be stimulated during surgical manipulation of the ocular and periocular structures (ophthalmic division of the trigeminal nerve), cranio-maxillofacial operations (ophthalmic, maxillary and mandibular divisions of the trigeminal nerve), and skull base surgery/neurosurgery (trigeminal ganglion) [3]. The incidence of TCR varies significantly with regards to the type of surgical procedure being performed. Ophthalmic surgeries and surgeries involving manipulation of the orbit have a 90% incidence of OCR, a subtype of TCR [4]. Cranio-maxillofacial surgery including the LeFort-I-osteotomy; midface fracture reduction; temporomandibular joint arthroscopy/surgery; zygomatic arch elevation; and manipulation of soft tissues innervated by the mandibular, maxillary, and ophthalmic divisions of the trigeminal nerve have a 1–2% incidence of TCR [4]. Skull base surgery including microvascular decompression surgery of the trigeminal nerve in the cerebellopontine angle, balloon-compression rhizotomy of the trigeminal ganglion, and transsphenoidal pituitary surgery have an 8–18% incidence of TCR [4]. Although most documented cases of TCR have been intra-operative occurrences, there have been reports of delayed TCR following maxillofacial surgery in the post-operative period. A patient who underwent an uneventful zygomatic-orbital complex surgery and nasal bone fracture reduction with placement of nasal packings bilaterally was transferred to the Post-Anesthesia Care Unit (PACU) where he developed dyspnea, bradycardia, hypotension, and hypoxemia [5]. After no improvement with 100% O_2_ administration, atropine was administered, which corrected only the bradycardia. TCR was suspected, and the bilateral nasal packings placed in the operating room were removed, with complete resolution of signs and symptoms within 5 minutes [5].

TCR poses a unique phenomenon for anesthesia providers as management and treatment options are varied. Risk factors for TCR include light anesthesia, hypercapnia, hypoxemia, as well as certain medications [6]. Minimizing these intraoperative risk factors is beneficial. Depth of anesthesia is an important consideration. One study found a greater than 4-fold pooled risk of asystole during TCR under light anesthesia versus deep anesthesia during skull-based surgery [7]. Different types of anesthetics have been shown to influence TCR. TCR was less often seen with sevoflurane than halothane in children undergoing strabismus correction [8]. Fewer hemodynamic changes induced by the oculocardiac reflex, a subtype of TCR, were seen with ketamine compared to propofol whether for induction or maintenance [9,10]. Keeping a deep anesthetic and blunting autonomic reflexes, as well as intense monitoring for bradycardia, hypoventilation, and hypercapnia may lessen the incidence of TCR [6,8].

In addition, medications including beta blockers, calcium channel blockers, and narcotics are known to increase a patient’s risk of TCR. Prophylactic options include vagolytic agents or peripheral nerve blocks. If TCR occurs, communication between the surgeons and anesthesiologist is paramount as cessation of surgical stimulus, traction or manipulation of the nerve should take place immediately. If unsuccessful in alleviating TCR, then vagolytic medications should be administered [11]. Atropine has been used as pre-treatment and treatment for TCR [12,13]. Adrenaline and glycopyrrolate are other valid options [11,12]. Some suggest that external pacemakers be reserved for elderly, high risk cardiac patients undergoing surgeries with risk of TCR, as there is concern atropine results in tachycardia or arrhythmias that can damage myocardium [14,15].

This paper’s contribution highlights TCR occurring in sphenopalatine ganglion implantation procedures due to stimulation of the maxillary division of the trigeminal nerve. The sphenopalatine ganglia (pterygopalatine ganglia) exist as a bilateral pair and are located in the pterygopalatine fossae. The sphenopalatine ganglion contains dual innervation via the facial and trigeminal nerves. The Vidian nerve or the nerve of the pterygoid canal is formed from the merging of the greater petrosal nerve (a parasympathetic branch of the facial nerve) and the deep petrosal nerve (sympathetic fibers). The post-ganglionic parasympathetic fibers travel through the ophthalmic and maxillary divisions of the trigeminal nerve. The maxillary division of the trigeminal nerve provides somatosensory nerve fibers that pass through the sphenopalatine ganglion and form the greater and lesser palatine nerves. A total of eight patients underwent sphenopalatine ganglion implantation of a neuromodulator device for intractable cluster headaches. A few of the first patients to undergo sphenopalatine ganglion implantation experienced profound TCR, cases 1–3, discussed below. Due to concern raised over these occurrences, pre-anesthesia external pacemaker pads were placed on all future patients undergoing the same surgery. Our contribution is a case series showing a unique condition in OMFS related to TCR.

## 2. Sphenopalatine Ganglion Implantation Procedure

The Pathway CH-2 clinical trial “Sphenopalatine Ganglion Stimulation for Treatment of Chronic Cluster Headache” was an FDA approved study sponsored by Autonomic Technologies, Inc. (ATI), commencing in 2014. It was a multi-center, interventional, randomized, placebo-controlled, parallel, triple-blind safety and efficacy study that included intra-operative procedures for the implantation of the ATI-Neurostimulator in the pterygopalatine fossa so that an active electrode was within 0.5 mm of the sphenopalatine ganglion to intercept cluster headache symptoms at the onset. Each patient signed informed consent approved by the UTSW IRB committee (STU#052015-044). The results of this clinical trial have been previously published [16]. The sphenopalatine ganglion is located at the terminal end of the vidian canal, which contains the greater and lesser petrosal nerves that are preganglionic parasympathetic afferents synapsing in the sphenopalatine ganglion, which are shown to be involved in inducing the trigeminal autonomic reflex onset of cluster headaches [17]. Thus, stimulating the sphenopalatine ganglion would intercept this interaction and terminate the cluster headache. Figure 1 shows a coronal CT scan and adjacent illustration showing the position of the vidian canal (VC) in the medial portion of the pterygopalatine fossa. Just lateral to the VC is the foramen rotundum (FR), which contains the main trunk of the trigeminal maxillary division (V2). The CT view shows visible electrodes properly positioned to stimulate the sphenopalatine ganglion, and the illustration shows the trajectory of the implant close to FR and the transmitter connection on the lateral maxilla. Implantation of the ATI-Neurostimulator is performed through a small surgical incision through the maxillary posterior mucosa to expose the base of the zygoma and the pterygomaxillary plate. An introducer is used to position the implant at the sphenopalatine ganglion site under fluoroscopy. Figure 2 shows an illustration of the introducer device used to position the ATI-Neurostimulator. The electrodes are inactive during placement and only stimulated after the implant has been fixated. The TCR events precede activation, thus, the proposed mechanism of TCR induction is a mechanical interaction with V2 near the FR during use of the introducer device and subsequent positioning of the implant for fixation. Figure 3 shows a lateral CT view of an ATI-Neurostimulator after fixation.

## 3. Cases

### 3.1. Case 1

A 48-year-old female with past medical history (PMHx) of smoking presented for a left sphenopalatine ganglion implant for intractable chronic cluster headaches. Her home medications included melatonin and sumatriptan. Her pre-op vitals were stable. For pre-induction, she received midazolam (2 mg), followed by induction with remifentanil (150 mcg), lidocaine (50 mg), propofol (140 mg), and rocuronium (30 mg) for induction. Maintenance of anesthesia included a remifentanil infusion at 0.1–0.125 mcg/kg/min combined with desflurane. Soon after incision, the patient experienced an episode of bradycardia (45 bpm) and hypotension (88/51 mmHg). Surgical stimulation was stopped with recovery of both bradycardia and hypotension 2 minutes later. Throughout the case, the patient’s oxygen saturation and end tidal carbon dioxide was maintained appropriately (SpO_2_ > 97%, EtCO2 30–40 mmHg). Post operatively her vital signs were stable. She was given sumatriptan in the PACU for a cluster headache. This patient did not possess risk factors for TCR other than maxillary nerve proximity and manipulation. Of note, this patient returned for explantation of her sphenopalatine ganglion implant 3 years later due to paresthesias, malocclusion, and persistent cluster headaches. It was removed without intraoperative complication.

### 3.2. Case 2

A 68-year-old male with PMHx of cluster headaches, hyperlipidemia, hypertension, right bundle branch block, and saddle embolus (2013) status post 6 months of Xarelto, presented for right sphenopalatine ganglion implant placement. His home medications included aspirin, metoprolol, telmisartan, and zolmitriptan. Pre-op vitals were stable. The patient was induced with remifentanil (50 mcg), lidocaine (50 mg), propofol (150 mg), and rocuronium (100 mg). His intubation was complicated by a grade 4 view upon initial direct laryngoscopy attempt, followed by a second successful attempt with a grade 2 view with the glidescope. Maintenance of anesthesia was accomplished with a remifentanil infusion at 0.1–0.125 mcg/kg/min along with desflurane. Intraoperatively, the patient experienced two episodes of bradycardia. The first episode of bradycardia (38 bpm) occurred during introduction of the introducer device into the pterygopalatine fossa. The episode resolved after the device was removed. There was reoccurrence 13 minutes later (34 bpm) during reinsertion of the device. Both episodes occurred before stimulation of the fixed device suggesting the cause for the TCR events was mechanical in origin. No pharmacologic intervention was used. Throughout the case, the patient’s oxygen saturation and end tidal carbon dioxide were maintained appropriately (SpO_2_ > 95%, EtCO2 30–36 mmHg). Post-operative vitals were stable. One risk factor for TCR was his home beta blocker, otherwise there was no evidence of light anesthesia, hypercapnia, and hypoxemia.

### 3.3. Case 3

A 55-year-old female with PMHx of asthma, diabetes, hyperlipidemia, seasonal allergies, and smoking, presented for left sphenopalatine ganglion implant placement. Her home medications included atorvastatin, gabapentin, glyburide, montelukast, nortriptyline, rizatriptan, and topiramate. Pre-op vitals were stable. For induction, she received remifentanil (100 mcg), lidocaine (50 mg), propofol (130 mg), and rocuronium (30 mg) for induction with a remifentanil infusion at 0.1–0.15 mcg/kg/min combined with desflurane for maintenance. The patient experienced an episode of bradycardia (41 bmp) upon insertion of the introducer device. She was treated with glycopyrrolate (0.2 mg) with recovery and adjustment of the introducer device. The patient’s oxygen saturation and end tidal carbon dioxide were maintained appropriately (SpO_2_ > 98%, EtCO2 31–34 mmHg). Post-operative vitals were stable. The patient had no risk factors for TCR other than mechanical sources related to the surgery.

### 3.4. Case 4

A 66-year-old female with PMHx of anxiety, cluster headaches, depression, diverticulitis, gallbladder disease, hepatitis, and reflux, presented for right sphenopalatine ganglion implant placement. Her home medications included aspirin, sumatriptan, topiramate, and verapamil. Pre-operative vital signs were stable. For pre-induction, she received midazolam (2 mg), followed by induction with remifentanil (100 mcg), lidocaine (50 mg), propofol (50 mg), and rocuronium (30 mg). Anesthesia was maintained with remifentanil boluses of 39.375 mcg every 5 min combined with desflurane. There were no intraoperative events concerning for TCR.

### 3.5. Case 5

A 47-year-old female with PMHx of cluster headaches, hypothyroidism, and kidney stones, presented for left sphenopalatine ganglion implant placement. Her home medications included excedrine, levothyroxine, lithium carbonate, sumatriptan, trazadone, and verapamil. Pre-operative vital signs were stable. For pre-induction, she received midazolam (2 mg), followed by induction with remifentanil (100 mcg), lidocaine (50 mg), propofol (200 mg), and rocuronium (50 mg). Maintenance of anesthesia included a remifentanil infusion at 0.1–0.125 mcg/kg/min combined with desflurane. No events concerning TCR occurred.

### 3.6. Case 6

A 46-year-old male with PMHx of anxiety, cluster headaches, chronic right shoulder pain, reflux, and skin cancer, presented for left sphenopalatine ganglion implant placement. His home medications included diazepam and verapamil. Pre-operative vital signs were stable. For pre-induction, he received midazolam (2 mg). For induction, he received remifentanil (100 mcg), lidocaine (50 mg), propofol (150 mg), and rocuronium (50 mg). Maintenance of anesthesia included a remifentanil infusion at 0.15 mcg/kg/min combined with desflurane. There were no intraoperative events concerning TCR.

### 3.7. Case 7

A 70-year-old female with PMHx of cluster headaches, hypertension, and hypothyroidism, presented for right sphenopalatine ganglion implant. Home medications included levothyroxine, olmesartan, and verapamil. Pre-op vital signs were stable. For pre-induction, she received midazolam (2 mg), followed by induction with remifentanil (50 mcg), lidocaine (50 mg), propofol (120 mg), and rocuronium (40 mg). Anesthesia was maintained with a remifentanil infusion at 0.1–0.15 mcg/kg/min combined with desflurane for maintenance. No events concerning TCR occurred.

### 3.8. Case 8

A 33-year-old male with PMHx of depression, seizures, diverticulitis, rhabdomyolysis secondary to statin use, and cluster headaches, presented for left sphenopalatine ganglion implant placement. His home medications included gabapentin, lithium carbonate, melatonin daily, sumatriptan, testosterone, topiramate, and verapamil. Vital signs pre-operatively were stable. For pre-induction, he received midazolam (2 mg), followed by induction with remifentanil (150 mcg), lidocaine (50 mg), propofol (150 mg), and rocuronium (40 mg). Maintenance of anesthesia included a remifentanil infusion at 0.1–0.125 mcg/kg/min combined with desflurane. There were no intraoperative events concerning for TCR.

## 4. Discussion

Knowledge of the mechanism and management of the trigeminocardiac reflex (TCR) is important for the cranio-maxillofacial surgeon and anesthesia provider. Identification of procedures with a higher risk for TCR occurrence is important in order to ensure there is clear communication between the surgical team and anesthesia provider of the possible need for intervention. Pre-operatively, the surgeon should inform the anesthesia provider of the possibility of TCR when traction or manipulation of the ophthalmic, maxillary, and/or mandibular divisions of the trigeminal nerve is expected as can be seen during osteotomies of the maxilla/mandible, midface fracture reduction, temporomandibular joint arthroscopy/surgery, zygomatic arch elevation, sphenopalatine ganglion implantation, and manipulation of the soft tissues of the face. Intra-operatively, the surgeon should use gentle manipulation of the peripheral branches of the trigeminal nerve. In addition, intra-operative communication between the surgical team and anesthesia provider should be clear especially when the surgeon is approaching or expecting to manipulate a division of the trigeminal nerve. When TCR does occur, it is important to identify this as quickly as possible so that the stimulus can be withdrawn immediately, which will often lead to resolution of the hemodynamic changes that occur from TCR. If withdrawal of the stimulus does not lead to resolution of TCR, then an anticholinergic (atropine or glycopyrrolate) should be administered intravenously. Lastly, chest compressions or external pacing may be considered.

A preventative approach should be taken to minimize the risk factors that potentiate TCR events. In the pre-operative setting, knowing the patient’s history; co-morbid conditions, which many contribute to hypercapnia or hypoxemia intraoperatively; as well as home medications paying close attention to calcium channel blockers, beta blocker, and narcotic use is imperative [6,11]. Only one of the patients with TCR events was taking a beta blocker pre-operatively. None were opioid users. These TCR events were attributed to surgical manipulation and stimulation of the nerve as there were minimal to no other predicting factors identified and the TCR events stopped with removal of the surgical stimulus.

Intraoperatively, it is paramount to minimize hypoxic and hypercarbic events. An appropriate depth of anesthesia should be maintained as light anesthesia can potentiate TCR. In our case series, adequate oxygenation and ventilation were maintained. There was no concern for intraoperative hypoxemia or hypercarbia precipitating the TCR events [6]. In addition, depth of anesthesia was appropriate. In our case series, the anesthetic combination was not changed from remifentanil and desflurane to an alternative as this is a retrospective study. In two cases, TCR events stopped after surgical manipulation cessation. In one case, there was resolution of the TCR event with cessation of the surgical stimulus and anticholinergic treatment. Laterality of sphenopalatine ganglion did not appear to play a role as the TCR events occurs in patients undergoing sphenopalatine ganglion implantation on both the left and right sides. Overall, none of the surgical cases were cancelled due to TCR events and patients who experienced TCR events intraoperatively did not demonstrate adverse side effects post-operatively.

During sphenopalatine ganglion implantation, the anesthesia provider must be cognizant of a possible TCR event and be prepared to treat with an anticholinergic medication. To the authors’ knowledge, no clinical trials or studies definitively state whether external pacing pads should be placed at the start of surgeries with a risk for TCR events. We believe pre-operative placement of external pacing pads in high risk cardiac patients may be beneficial in sphenopalatine ganglion implantation due to the proximity of the vidian nerve and maxillary nerve during implant placement. While this case series did not have adverse outcomes secondary to cardiac complications, it is up to the patient’s team to consider the risk and benefit of external pacing pads.

The contribution of this case series is to suggest a novel condition in OMFS related to TCR. The source of TCR is stimulation of the maxillary division of the trigeminal nerve and is not due to the sphenopalatine ganglion. This is supported by the fact that none of the post-implantation stimulated patients experienced TCR, and the TCR events only occurred during the actual implantation procedure. Intentional stimulation of the sphenopalatine ganglion without TCR demonstrates that the maxillary nerve was the source for TCR. Appropriate pre-operative discussion between the surgical team and anesthesia provider is important to minimize potential risk factors. Communication during sphenopalatine ganglion implantation is paramount, especially when the surgeons may be near or at risk of manipulating or stimulating of the maxillary nerve. This will allow the team to be prepared and ready to either pre-treat in advance or treat if TCR occurs. The anesthesia provider must be vigilant to ensure appropriate depth of anesthesia, cardiovascular monitoring, and appropriate oxygenation and ventilation. Future studies may explore utilizing alternative maintenance anesthetics and its effect on TCR events. There is limited data with respect to pre-anesthesia placement of external pacing pads. This may be beneficial in high-risk cardiac patients undergoing OMFS/CFS surgeries that have a high likelihood of TCR events. Future studies are warranted to provide a clearer answer.

## Figures and Tables

**Figure 1 brainsci-10-00973-f001:**
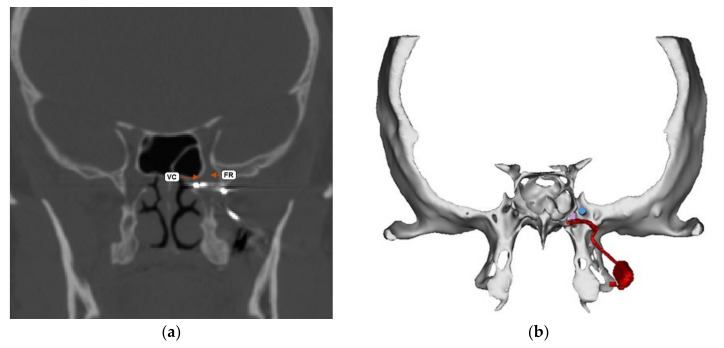
Coronal CT and illustration of the ATI-Neurostimulator in the pterygopalatine fossa with the terminal electrode within 0.5 mm of the sphenopalatine ganglion at the vidian canal (VC). The location of the foramen rotundum (FR) is shown in close proximity and lateral to the VC. (**a**) Shows CT image and terminal electrodes in proper position. (**b**) Shows illustration of A showing the position of the electrodes and the stimulators components on the lateral posterior maxilla.

**Figure 2 brainsci-10-00973-f002:**
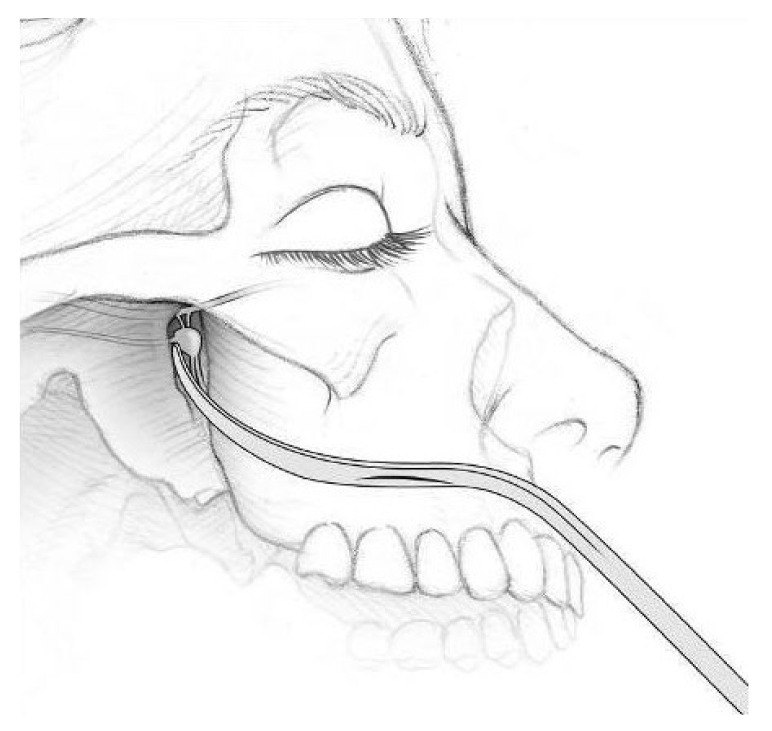
An illustration showing the trajectory of the implant introducer device used to place the ATI-Neurostimulator at the location of the sphenopalatine ganglion in the pterygopalatine fossa from an intra-oral incision at the posterior superior vestibule of the maxilla.

**Figure 3 brainsci-10-00973-f003:**
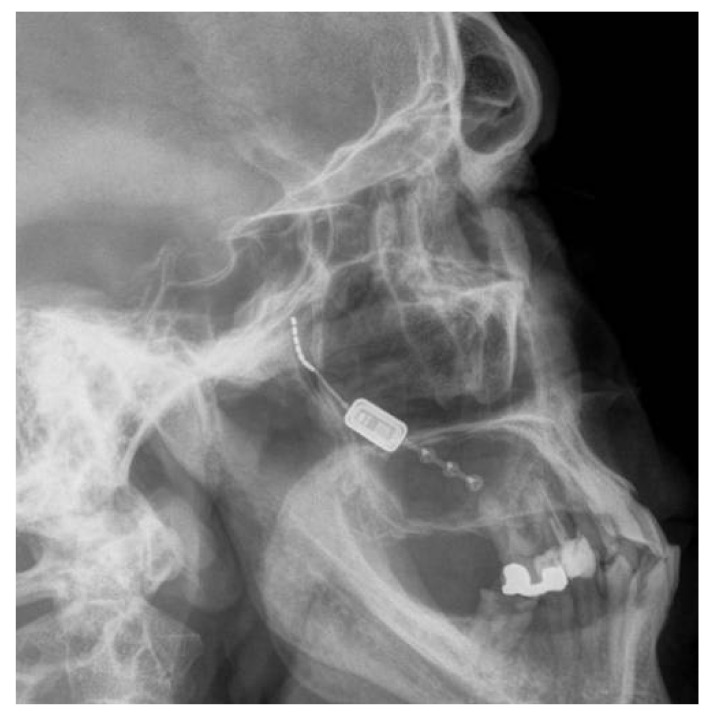
A lateral radiograph showing the proper position of an ATI-Neurostimulator with electrodes within the pterygopalatine fossa and fixation to the posterior maxillary buttress.

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
