# Peer review of "Trigeminocardiac Reflex Induced by Maxillary Nerve Stimulation during Sphenopalatine Ganglion Implantation: A Case Series"

_brainsci, 2020, doi:10.3390/brainsci10120973_

Round 1
Reviewer 1 Report
The Authors present a small case series of 8 patients who underwent maxillo-facial surgery for implantation of sphenopalatine ganglion stimulator, which has been recently shown as a new promising therapy for treatment-resistant chronic cluster headache.The case series is extracted from a large dataset of 45 patients, addressing efficacy and safety of this novel therapeutic option. In this case series, the Authors highlight the occurrence of trigeminocardiac reflex causing reversible bradycardia and hypotension during the surgical procedure, experienced by three of the eight patients. This aspect has been neglected by the previous paper, probably because it did never require the cancelation of surgery and never provoke any post-procedural side effects on the patient. However, this report is of great importance to the surgeons and anesthesiologists who need to know in advance which may be the intraoperative complications, their risk factors and how to promptly manage them. I have some observation:
1)The absence of need of surgery cancelation, and the absence of post-procedural side effects in all the patients who experienced trigeminocardiac reflex can be deduced by the previous paper, but must be stressed more clearly in this paper; 2) I would avoid the use of acronyms in the title. Please change "SPG" in "sphenopalatine ganglion" 3)I would remove the part "Requires anesthesia/surgery team management" from the title because ALL the surgical procedure would require the best possible anesthesia/surgery team management. The peculiarity of trigeminocardiac reflex is that a complication primary noted by the anesthesiologist (i.e. bradycardia and hypotension) requires an immediate intervention by the surgeon (the prompt arrest of nerve manipulation), but this point is adequately highlighted in the discussion. I suggest as title: "Occurrence of trigeminocardiac reflex causing reversible bradycardia and hypotension during sphenopalatine ganglion implantation: a case series" 4)In the introduction should be stressed that the sphenopalatine ganglion has a double innervation, from the facial and the trigeminal nerve, with very different functions. The side effect discussed in the article depends on the trigeminal innervation, while the therapeutic effect of nerve stimulation is believed to rely on facial innervation: may this provide a clue to the surgeon on how avoid occurrence of trigeminocardiac reflex or not? Please answer in the discussion.Author Response
Please see attached.

Reviewer 2 Report
Why is content 102-126 repeated in 127-151?
case 1 : versed is unknown , better use midazolam or versed midazolam
180 : malocculsion = malocclusion
Why is , after 3 cases where TCR occured , never changed to another anesthesia without desflurane and remifentanil?
why is there no use of sufentanil or fentanyl instead of remifentaniland IV/ TCI propofol anesthesia instead of desflurane
